# Parental Stressors in Sports Influenced by Attributes of Parents and Their Children

**DOI:** 10.3390/ijerph19138015

**Published:** 2022-06-30

**Authors:** Krisztina Kovács, Rita F. Földi, Gábor Géczi, Noémi Gyömbér

**Affiliations:** 1Department of Psychology and Sport Psychology, Hungarian University of Sports Science, 1123 Budapest, Hungary; gyomber.noemi@tf.hu; 2Department Developmental and Educational Psychology, Károli Gáspár University of the Reformed Church in Hungary, 1037 Budapest, Hungary; foldi.rita@kre.hu; 3Department of Sport Management, Hungarian University of Sports Science, 1123 Budapest, Hungary; geczi.gabor@tf.hu

**Keywords:** youth sport, parental stressors, parental involvement, injury

## Abstract

(1) Background: Although parental behavior is known to be an important source of influence, relatively few studies have examined the factors possibly contributing parental stressors as being directly related to their children’s sport socialization. The present study explored the relative importance of related parental stressors and the associations between these stressors and various types of parental involvement. (2) Method: A total of 1260 parents completed an online form including demographic questions, questions on their children’s sport participation, and three self-report measures (PSSS, PISQ, and PASSES). (3) Results: The results revealed that the multiple linear regression models for overall parental stress were statistically significant. The significant predictors were the parent’s educational level, the child’s current stage of sports participation, and the child’s sport injury (or the lack thereof). Furthermore, the stressors perceived by parents were positively associated with parental direct behavior and pressure. (4) Conclusions: Parents are under more stress as their child’s stages of sport development increases, if their child has already had a sports injury, and the parents’ directive behavior and experienced stress were significantly associated. The obtained results expand the existing knowledge of the complexity of parents’ importance in children’s sports careers.

## 1. Introduction

Stress is a much-debated topic; thus, various definitions exist. For instance, Selye [1] (p. 137) defined stress as “the nonspecific response of the body to any demand made upon it,” in which the non-specific response is part of the so-called general adaptation syndrome. In this model, the two major factors concerning the stress response is the cause (i.e., environmental stressors or stimuli, such as important life events or everyday nuisances) and the consequence (i.e., the subjective response elicited by the stressor, including the appraisal process and the coping response). Finally, Lazarus and Folkman [2] proposed a transactional model, which defines stress as a relationship between the individual and their environment and assigns decisive importance to the individual’s perception of the situation in terms of the emergence of stress. In this, the core of the model is the cognitive appraisal process which centers around the available coping resources and the potential threats.

According to the bioecological theory of human development proposed by Bronfenbrenner [3,4], parents are part of their child’s microsystem, that is, they have direct impact on their child’s sport career, while their parental attitudes and behavior may be substantially influenced by other systems (e.g., expectations at the workplace, or their relationship with the child’s sports club). Because of social and political impacts (i.e., politics, economical, and technological) youth sport is changing for participants (i.e., athletes, parents, coaches), where the expectations related to performance and competition turn out to be greater [5,6]. Several studies have found that not only coaches but also parents play a pivotal role in the development of athletes’ mindsets as the relationship between athletes and their parents’ attitudes towards sport [7,8,9], goal orientation [10,11], motivation [12] and perfectionism [13] were detected. Studies have also emphasized the stressful nature of sports parenting and the negative effect this can have on their experiences [14,15,16,17,18]. According to qualitative and quantitative studies, sporting children’s parents face different stressors compared to other parents. These unique stressors include higher time pressures [19], managing the sport-education balance [20], deselection [21,22], and concerns about the exclusive dominance of the sport in the child’s future career [17]. The related parental experiences vary by the child’s level of participation in sport [23], while the stressors affecting young athletes and their environment vary across age groups [17]. Harwood, Drew, and Knight’s qualitative research [17] revealed the following types of stressors in a sample of parents of specialization-age football academy players, namely, processes managed by the academy and the quality of their communication, competition-related stressors, conflicts between sport-related and family roles, and problems related to the child’s education. These stressors continue to affect parents in the investment stage, which Harwood and Knight [16] conceptualized into three types of stressors, namely, organizational influence and problems (e.g., logistical issues, financial support, and conflicts with the child’s sports club), competition-related challenges (e.g., dealing with the child’s failures and taking account of other parents’ behavior), and problems related to the child’s development (e.g., keeping contact with the school and prospects for a career after sport). Sutcliffe, Kelly, and Vella [19] noted that parents whose children pursue a career in sport experienced higher time pressures by those. Furthermore, based on Clarke and Harwood’s qualitative research [15], the burden of responsibility on parents was found to increase over the course of the child’s sports socialization, in addition to developing a sense of uncertainty and fear about possibly hindering their child’s sports career by presenting themselves in an unfavorable light in front of the sports academy.

The severity of stress experienced by parents is related to the characteristics of their involvement in their children’s sports careers. Relatively high levels of anxiety may result in increased parental pressure [24,25], which potentially contributes to negative emotional outcomes in the athletes, such as reduced pleasure in the sport, low self-confidence, and increased stress [26], while these, in turn, may eventually lead to early dropout [27,28,29,30,31]. Danioni and Barni [32] primarily assigned a control function to parental pressure in sport, of which children may have either positive or negative experiences depending on the specific situation.

The present study explored the relative importance of stressors potentially influencing the parenting behavior of junior athletes’ parents and the associations between these stressors and various types of parental involvement. The findings contribute to the expanding knowledge of parents’ complex influence on their children’s sports career.

## 2. Materials and Methods

### 2.1. Procedure

The participants were recruited via convenience sampling. The participants were contacted directly or with the help of sports associations, sports clubs, a sports academy, and professional coaches. All these contributors and all participants gave consent to participate after being informed of the purpose of the study, the employed instruments, and the relevant ethical standards, including confidentiality. The participants completed an online form including demographic questions, questions on their children’s sports participation, and self-report psychological measures. The research plan was approved by the Research Ethics Committee of the University of Physical Education, Budapest, Hungary, under License No. TE-KEB/No1/2019. The statistical data analysis was carried out with the IBM SPSS Statistics v. 22 software package (IBM, Armonk, NY, USA).

### 2.2. Participants

The sample included a total of N = 1260 participating parents (399 males and 861 females) aged 29 to 74 years (Mage = 43.54, SD = 5.10). All participants had at least one child who regularly engaged in a team sport as a registered member of a sports club or sports academy at the time of data collection and 45.3% of them had themselves previously been athletes at a competitive level. Their children, for whom they provided further details, comprised 906 boys (71.9%) and 354 girls (28.1%) aged 5 to 21 years (M = 12.88, SD = 2.89). By the stage of sports participation, 31.4% of the children were at the sampling stage, 50.0% at the specializing stage, and 18.6% at the investment stage. The children engaged in handball (50.2%), football (34.8%), basketball (7.6%), water polo (3.5%), ice hockey (2.9%), and volleyball (1.0%).

### 2.3. Instruments

The participants provided data on both their own and their children’s sociodemographic and sport-relevant characteristics (parents’ gender, age, educational level, and past sporting experiences; children’s gender, age, sport, training hours per week, active years in the sport, sports participation stage, and past injuries).

#### 2.3.1. Parental Involvement in Sport Questionnaire (PISQ)

The 15-item Parental Involvement in Sport Questionnaire [33,34] consists of four subscales as follows. The Directive Behavior subscale measures the extent to which a parent strives to control her/his child’s sport-related behavior directly; the Active Involvement subscale assesses parental attendance of training sessions and competitions; the Praise and Understanding subscale taps the relative importance of empathetic and understanding parental behavior, while the Parental Pressure subscale provides a measure of the importance of parental expectations for the child’s good performance and success in competition. Each Likert item is rated on a five-point scale. The Hungarian version of the PISQ showed acceptable–good internal consistency (Cronbach’s α = 0.65 to 0.88).

#### 2.3.2. Perceived Autonomy Support Scale for Exercise Settings (PASSES-H)

The Hungarian version of the Perceived Autonomy Support Scale for Exercise Settings [35] was originally designed to assess junior athletes’ perceptions of parental autonomy support. For the purposes of the present study, both the instructions and the 12 original items were reworded to obtain a measure of parents’ self-perceived autonomy support provision (e.g., the original item My parents encourage me to do active sports and/or vigorous exercise in my free time was replaced with the item I encourage my child to do active sports and/or vigorous exercise). Each Likert item is rated on a seven-point scale ranging from Strongly disagree (1) to Strongly agree (7). Higher overall scores indicated higher levels of parental autonomy support. The scale showed good internal consistency (Cronbach’s α = 0.83).

#### 2.3.3. Parental Stressors in Sport Scale (PSSS)

The PSSS [36] assesses the subjective importance of several parental stressors directly related to the child’s sports career. The 11-item Likert scale comprises four subscales, while the total score indicates the anxiety currently experienced by the respondent in relation to their child’s sports career. The Deselection subscale reflects the fear of performance expectations and their negative consequences, while the Feedback subscale shows how well parents are informed by the club about their children’s development. The subscale of club-related stressors provides information on the effectiveness of the parent-club relationship, and the subscale of Education-related stressors reflects the fears of parents regarding the negative influence of competitive sport on school performance and educational development. Each Likert item is rated on a five-point scale. The PSSS showed acceptable–very good internal consistency (Cronbach’s α = 0.63 to 0.98).

### 2.4. Statistical Data Analysis

First, the obtained data were analyzed for possible mean differences on the five PSSS scales between groups based on variables tested in previous studies (the parent’s gender, the child’s gender, the parent’s previous sporting experience, the child’s sports participation stage as defined by Côte and Hay [37]) and on variables introduced in the present study (the child’s sports injuries, the parent’s marital status, the parent’s level of education, and the number of children in the family). Mean differences were tested with robust statistical methods, including Welch’s independent samples *t*-tests and Welch’s one-way ANOVA tests with Bonferroni-adjusted Games–Howell post hoc tests.

Second, the possible predictors of each parental stressors type were tested with linear regression analyses (Enter method). Taking account of the related previous findings, the same model was consistently tested for each of the five PSSS scales as the dependent variables, in which the parent’s educational level (high school/ university), the child’s sports participation stage (sampling/specializing/investment), and the child’s sports injuries were entered. For the purposes of linear regression analysis, the child’s sports participation stage was each recoded into two dummy variables. Finally, the associations between the employed psychological measures were tested with partial correlation coefficients, controlling for the parent’s gender, the parent’s previous sporting experience, the child’s age, and the child’s sports injuries.

## 3. Results

The means, standard deviations, minimum and maximum values, Cronbach’s αs, and confidence intervals of Cronbach’s αs that were obtained for the PISQ subscales are presented in Table 1.

### 3.1. Differences in Overall Parental Stress and the Relative Importance of Different Stressors

The results of the independent samples *t*-test and the one-way ANOVA test (with Bonferroni adjustment, *p* < 0.013) revealed that the parents of children at the sampling stage experienced significantly lower levels of stress, fear of deselection, and conflict with the sports club/academy compared to the parents of children at the investment or specialization stage. No significant difference was found between the parents of children at the investment vs. specialization stage (see Table 2).

The parents of children with a previous sports injury reported higher levels of fear of deselection and negative experiences related to communications from the club compared to the parents of children with no previous sports injury. Furthermore, the parents with tertiary education reported higher levels of concern about their children’s academic progress compared to those with secondary education (see Table 3). No significant differences on either PSSS subscale were associated with the parent’s gender, the child’s gender, the parent’s previous sporting experience (yes/no), the parent’s marital status (single/married/cohabitant), or the number of children in the household (one/two/three or more).

### 3.2. Predictions on Overall Parental Stress and on the Relative Importance of Different Stressors

Since the previous related studies did not reveal any predictors that explained substantial variance in parental stress, all predictors employed in the present study were entered in one step in each of five multiple linear regression models with overall parental stress and each parental stressor as the outcome variable. The entered predictors were the parent’s level of education (secondary/tertiary), the child’s sports participation stage (sampling/specialization/investment; entered as two dummy variables, encoding sampling and investment), and the child’s previous sports injury (yes/no).

The results revealed that the model for overall parental stress was statistically significant (*F*(4,1236) = 10.511, *p* < 0.001). The model explained 3.3% of the total variance in parental stressors. The significant predictors were the parent’s level of education, having a child at the sampling stage, and the child’s previous sports injury (see Table 4). Specifically, the parents with tertiary (vs. secondary) education and those with children have suffered a sports injury (vs. no injury) reported higher levels of stress, while those with children at the sampling stage (vs. either of the other two stages) reported lower levels of stress.

All four models for the four PSSS subscales were statistically significant (see Table 4). The model for deselection explained 1.7% of the total variance in the outcome variable (*F*(4,1236) = 5.261, *p* < 0.001). However, none of the employed predictors were significant.

The model for club-related stressors explained 1.6% of the total variance in the outcome variable (*F*(4,1236) = 5.020, *p* < 0.005). The significant predictors were the parent’s level of education and the child’s previous sports injury. Specifically, the parents with tertiary (vs. secondary) education and those with children have suffered a sports injury (vs. no injury) reported higher levels of club-related stress.

The model for inadequate feedback explained 3.1% of the total variance in the outcome variable (*F*(4,1236) = 9.991, *p* < 0.001). The significant predictors were the parent’s level of education and the child’s previous sports injury. Specifically, the parents with tertiary (vs. secondary) education and those with children having suffered a sports injury (vs. no injury) reported higher levels of feedback-related stress.

Finally, the model for education-related stressors explained 1.3% of the total variance in the outcome variable (*F*(4,1236) = 3.973, *p* < 0.005). The significant predictors were the parent’s level of education and having a child at the sampling stage. Specifically, the parents with tertiary (vs. secondary) education reported higher levels of stress, while those with children at the sampling stage (vs. either of the other two stages) reported lower levels of stress.

### 3.3. Associations between the Psychological Measures

The associations between the employed psychological measures were tested with partial correlation coefficients, controlling for the parent’s gender, the parent’s previous sporting experience, the child’s age, and the child’s sports injury. Both overall parental stress and deselection showed low but significant positive partial correlations with directive parental involvement and parental pressure, while both mentioned PSSS measures correlated negatively with autonomy support. Furthermore, education-related stressors showed a low but significant negative partial correlation with autonomy support (see Table 5).

## 4. Discussion

The present study explored the relative importance of stressors potentially influencing the quality of parental involvement in junior athletes’ sports careers, and it also tested possible predictors of parental stress related to various stressors. The obtained results revealed that the parent’s education, the child’s sports participation stage, and the child’s previous sports injury significantly predicted the parent’s stress related to several stressors.

The results of the present study show that the child’s sports participation stage could be a significant predictor of parental stress and confirm that their stress is lowest during the sampling years (under 12 years), while it substantially increases after the child reaches 12 years old when parental participation is significant in the child’s sports life. The related findings show that parents may also be influenced by external expectations and circumstances that significantly impact their athletic children. Sports parenting expertise is not confined to children’s emotional support. Parents manage their children’s sports-studies-leisure balance, mediate between various participants in their children’s life, fulfill logistic duties, provide financial support for their children’s sports career, and often make decisions that require sacrifices from the family [38,39]. While the importance of parental participation in children’s sports careers decreases over time [40,41], the number of related duties and responsibilities increases at each successive stage of sports participation [16,17], leading to significant stressors in their life. The roles of parents change over the different stages of athletic development [37,41]. In the specialization and investment years, as a result of the increasing burdens accompanied by decreasing parental involvement, parents may develop a sense of uncertainty about the previously anticipated return on the sacrifices they made for their children’s progress in the sport. The parents could feel responsible for managing their child’s stressful, insecure times. Stress might be caused by the pressure to make a “good decision” or manage their child’s tangible and intangible demands about their sports career [5]. One such issue could be the management of a dual career, that is, the decisions to be made on further education that should enable the child to pursue a career after quitting competitive sport from the specialization years [41]. Furthermore, pursuing a career in sport involves substantial risk; an injury, changing coaches or clubs, or a season with poor results may severely hinder or even prematurely end the child’s sports career [14,17,22]. Finally, due to athletic children’s strict schedule, their private life also requires careful planning to protect the quality of their peer relationships, leisure time, and family life [37,38,39]. All these decisions have to be made by the parents, who need to take account of many factors to meet their children’s needs and serve their best interests. Considering that the child’s education could be a significant parental stressor, especially at later stages of sports participation, it is important to increase the capacity of sports clubs, academies, and schools to support children in pursuing a dual career and ease parents’ burdens.

The elite sport involves intense competition and thus the risk of physical injury. The risk of sports injury emerges as a significant parental stressor at the specialization stage [16,17,42], in line with our findings. Parents are not only concerned about the injury itself but also about the difficulties they may have to face during the rehabilitation process. Beyond the difficulties of ensuring adequate health care, including rehabilitation experts and those keeping contact with the sports club, a severe injury may delay children in, or even prevent them from, achieving their sport-related goals. Questions may arise concerning the severity of the lag for the child to catch up with after resuming elite sport, how it affects their standing at the club, and how these circumstances affect their long-term goals. Identifying and securing the available forms and sources of post-injury social support may lay the ground for effective cooperation between parents, athletes, and practitioners, which in turn may improve the effectiveness of rehabilitation [43].

Only a few previous studies addressed the impact of parents’ level of education on their children’s sports careers. Findings obtained from a Japanese sample suggest that parents with relatively low education are more likely to tolerate coaches’ negative behavior and communication than those with a higher level of education [44]. This is in line with the findings of the present study in which the parents’ educational level might predict their level of stress. Our results indicate that the parents with secondary (i.e., relatively low) education may have found communication with coaches and the club less stressful than those with tertiary education, who were more concerned about their children’s academic progress and more likely to confront the club over its communication. These findings may be useful for both parents and sports professionals.

Hoover-Dempsey and Sandler [45] considered high levels of parental stress as a factor that, in fact, hinders children’s progress via its adverse impact on parental involvement. This idea was also supported by the findings of the present study, which revealed that parental stress was positively associated with directive parental behavior and parental pressure, and these associations were independent of the parent’s gender, the parent’s previous sporting experience, the child’s age, and the child’s previous sports injury. Further studies are needed to clarify the possible causal relationships between parental stress and the controlling forms of parental involvement, that is, whether elevated stress leads to stronger parental control or vice versa. Previous findings show that positive parental behavior is related to parental support for autonomy [46] and effective emotional self-regulation [47]. In line with these findings, the present study found a negative association between parental support for autonomy and parental stress; that is, not only junior athletes but parents themselves may enjoy better well-being by learning and exercising autonomy support.

None of the parent’s gender, the child’s gender, the parent’s previous sporting experience, the parent’s marital status, and the number of children in the household accounted for significant differences in parental stress. Although most related studies found the child’s age to be a factor influencing parental stress [17,22,39], most of these studies were aimed at identifying the major stressors or coping strategies [14,18,48]. The PSSS employed in the present study assesses state (vs. trait) anxiety, which is primarily determined by situational conditions rather than intra-individual factors, which is also reflected in the obtained findings.

The most important limitation of the present findings is that they are based on self-report data, which are liable to social desirability effects. In addition to self-report measures, future studies should rely on qualitative methods in exploring the stress-related psychological characteristics of junior athletes’ parents, which are otherwise more popular in research on parental behavior with junior athletes. Although the tested multiple linear regression proved significant for all types of parental stressors, the obtained coefficients of determination were small. Parenting is, of course, not confined to parents’ stress levels or involvement in their children’s sports careers, while the intensity of parental stressors itself may be influenced by a variety of factors. A useful interpretive framework is offered by Lazarus and Folkman’s [2] transactional model that defines stress as a relationship between the individual and their environment and assigns decisive importance to the individual’s perception of the situation in terms of the emergence of stress, and Bronfenbrenner’s ecological model [3,4], which points out the importance of the expectations held by the immediate environment and the norms prevailing in wider society in terms of parental stress. These factors fell outside the scope of the present study, while they might essentially contribute to the prediction of the intensity and types of parental subjective experience. The associations of parental stressors with parental involvement and parental autonomy support were tested with correlation coefficients, which did not reveal any information on the possible causal relationships between the involved variables; thus, these relationships remain the subject of further research. No socioeconomic measure was employed in the present study, while parents’ socioeconomic status may be an important factor in the intensity of stress experienced in relation to parental functions. Although the participants provided data on their marital status, which allowed for assigning them to the two broad categories of married/cohabitant and single, no further data were obtained on the characteristics of married/cohabitant participants’ intimate partner relationships, which may also influence the intensity of parental stress. The study did not establish whether the sample included participants who were simultaneously active as parents and coaches, possibly training their own children, while this dual role may clearly have an impact on the quality of parental involvement and the intensity of parental stress [49]. Considering the rare occurrence and special dynamics of a dual parent-coach role, qualitative methods may be particularly fruitful in related future studies.

## 5. Conclusions

The intensity of parental stress might be influenced by the child’s stages of sports development. Apparently, parental involvement could decrease while parental stress increases over time. Other important parental stressors might include conflicts with the child’s sports club or school and the risk of dropout. The importance of these stressors might presumably be related to the unpredictability of the child’s sports career, which is dependent on several factors outside junior athletes’ and their parents’ control, such as suffering a sports injury, for example. These findings have important implications for parental psychoeducation as well as for the training of sports professionals (coaches, sports medicine physicians, rehabilitation experts).

## Figures and Tables

**Table 1 ijerph-19-08015-t001:** Descriptive statistics and Cronbach alpha (N = 1260).

		Min.	Max.	M	SD	Cronbach α	95% Confidence Interval	
							Lower Bound	Upper Bound
PSSS	Parental Stressors Total Score	11	47	21.10	7.27	0.82	0.81	0.84
	Deselection	4	20	7.64	3.38	0.63	0.6	0.66
	Club-Related Stressors	3	12	5.87	2.36	0.74	0.72	0.77
	Feedback *	2	10	4.82	2.64	0.98 *	0.97	0.98
	Education-Related Stressors	2	10	2.77	1.47	0.86	0.85	0.88
PISQ	Directive Behavior	6	30	14.56	5.85	0.88	0.87	0.89
	Parental Pressure	4	20	6.70	2.96	0.74	0.71	0.76
	Active Involvement	2	10	3.94	2.26	0.78	0.76	0.80
	Praise and Understanding	5	15	13.93	1.58	0.65	0.62	0.69
Perceived Parental Autonomy Support		35	84	76.29	7.07	0.83	0.82	0.85

Note. * the subscale contains two items; Alpha values: acceptable: 60 < α < 0.70; good: 0.70 < α < 0.90 (Taber, 2018).

**Table 2 ijerph-19-08015-t002:** Differences in parental stress by children’s sport participation stage.

		N	M	SD	*F*	*p*	Games–Howell Post Hoc
Parental Stressors Total Score	Sampling	396	19.77	6.86	12.728	<0.001	Sa < Sp; Sa < I; I = Sp
	Specialization	630	21.34	7.21			
	Investment	234	22.70	7.74			
	Total	1260	21.10	7.27			
Deselection	Sampling	396	7.14	3.10	8.846	<0.001	Sa < Sp; Sa < I; I = Sp
	Specialization	630	7.71	3.38			
	Investment	234	8.30	3.70			
	Total	1260	7.64	3.38			
Club-Related Stressors	Sampling	396	5.65	2.32	3.730	0.025	
	Specialization	630	5.90	2.38			
	Investment	234	6.18	2.35			
	Total	1260	5.87	2.36			
Feedback	Sampling	396	4.37	2.50	11.362	<0.001	Sa < Sp; Sa < I; I = Sp
	Specialization	630	4.89	2.66			
	Investment	234	5.37	2.71			
	Total	1260	4.82	2.64			
Education-Related Stressors	Sampling	396	2.60	1.30	4.222	0.015	
	Specialization	630	2.84	1.51			
	Investment	234	2.86	1.60			
	Total	1260	2.77	1.47			

**Table 3 ijerph-19-08015-t003:** Differences in parental stress by children’s sports injury (vs. non-injury) and parents’ level of education.

			N	M	SD	*t*	*p*
Child’s Sport Injury	Parental Stressors Total Score	Yes	531	22.31	7.52	5.038	<0.001
		No	729	2.22	6.96		
	Deselection	Yes	531	8.02	3.58	3.357	0.001
		No	729	7.37	3.20		
	Club-Related Stressors	Yes	531	6.17	2.38	3.808	<0.001
		No	729	5.66	2.32		
	Feedback	Yes	531	5.25	2.76	4.885	<0.001
		No	729	4.51	2.50		
	Education-Related Stressors	Yes	531	2.88	1.58	2.191	0.029
		No	729	2.69	1.37		
Parent’s Level of Education	Parental Stressors Total Score	Secondary	557	2.52	6.97	−2.710	0.007
		Tertiary	684	21.64	7.49		
	Deselection	Secondary	557	7.56	3.31	−0.793	0.428
		Tertiary	684	7.72	3.45		
	Club-Related Stressors	Secondary	557	5.72	2.31	−2.265	0.024
		Tertiary	684	6.02	2.39		
	Feedback	Secondary	557	4.60	2.61	−2.879	0.004
		Tertiary	684	5.03	2.66		
	Education-Related Stressors	Secondary	557	2.65	1.31	−2.761	0.006
		Tertiary	684	2.87	1.58		

**Table 4 ijerph-19-08015-t004:** Hierarchical linear regression models for the parental stressors scale.

		B	SE	β	*t*	*p*			B	SE	β	*t*	*p*
Parental Stressors Total Score	Sampling ^a^	−1.030	.485	−0.066	−2.123	0.034	Fear of Deselection	Sampling ^a^	−0.432	0.227	−0.059	−1.897	0.058
	Investment ^a^	0.981	0.565	0.052	1.735	0.083		Investment ^a^	0.489	0.265	0.056	1.845	0.065
	Sport Injury	1.466	0.452	0.100	3.244	0.001		Sport Injury	0.383	0.212	0.056	1.808	0.071
	Education	1.156	0.409	0.079	2.823	0.005		Education	0.164	0.192	0.024	0.853	0.394
*R*^2^ = 0.033, *R*^2^_adj_ = 0.030, *F*(4,1236) = 1.511, *p* < 0.001							*R*^2^ = 0.017, *R*^2^_adj_ = 0.014, *F*(4,1236) = 5.261, *p* < 0.001						
Club- Related Stressors	Sampling ^a^	−0.079	0.158	−0.016	−0.501	0.617	Feedback	Sampling ^a^	−0.325	0.176	−0.057	−1.841	0.066
	Investment ^a^	0.188	0.185	0.031	1.020	0.308		Investment ^a^	0.313	0.206	0.046	1.524	0.128
	Sport Injury	0.418	0.148	0.088	2.835	0.005		Sport Injury	0.549	0.164	0.103	3.341	0.001
	Education	0.312	0.134	0.066	2.335	0.020		Education	0.447	0.149	0.084	3.005	0.003
*R*^2^ = 0.016, *R*^2^_adj_ = 0.013, *F*(4,1236) = 5.020, *p* = 0.001							*R*^2^ = 0.031, *R*^2^_adj_ = 0.028, *F*(4,1236) = 9.991, *p* < 0.001						
Education-Related Stressors	Sampling	−0.194	0.099	−0.061	−1.960	0.050							
	Investment	−0.009	0.115	−0.003	−0.082	0.935							
	Sport Injury	0.116	0.092	0.039	1.252	0.211							
	Education	0.233	0.084	0.079	2.781	0.006							
*R*^2^ = 0.013, *R*^2^_adj_ = 0.010, *F*(4,1236) = 3.973, *p* = 0.003													

^a^ Predictor: Child’s sport participation stage (dummy variable; excluded category: specialization).

**Table 5 ijerph-19-08015-t005:** Correlations between parental involvement and parental stressors.

		PSSS				
		Parental Stressors Total Score	Deselection	Club-Related Stressors	Feedback	Education- Related Stressors
PISQ	Directive Behavior	0.205 (0.228) **	0.233 (0.264) **	0.133 (0.149) **	0.137 (0.145) **	0.019 (0.018)
	Pressure	0.258 (0.261) **	0.300 (0.308) **	0.181 (0.183) **	0.136 (0.131) **	0.052 (0.049)
	Active Involvement	−0.148 (−0.146) **	−0.121 (−0.115) **	−0.046 (−0.042)	−0.138 (−0.138) **	−0.133 (−0.133) **
	Praise and Understanding	−0.129 (−0.134) **	−0.139 (−0.134) **	−0.059 (−0.056) *	−0.059 (−0.045) *	−0.117 (−0.107) **
Perceived Parental Autonomy Support		−0.210 (−0.225) **	−0.133 (−0.147) **	−0.189 (−0.200) **	−0.115 (−0.126) **	−0.221 (−0.222) **

Partial correlations in parentheses, ** *p* < 0.001, * *p* < 0.05.

## Data Availability

The data presented in this study are available on request from the corresponding author.

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
