# Peer review of "Parental Stressors in Sports Influenced by Attributes of Parents and Their Children"

_ijerph, 2022, doi:10.3390/ijerph19138015_

Round 1
Reviewer 1 Report
The aim of the study was to explore the importance of certain parental stressors and the relationship of these stressors with parental involvement in their children's sports activities. The study, which is correlational, was conducted with a large sample of 1260 parents who responded to various self-report instruments.
I believe that the study may be of interest to readers of the journal. However, I think that the following aspects should be modified in the manuscript.
1) The values of the reliability coefficients of the instruments used (lines 119, 130, and 142) should correspond to the data of the study, not to the data published in the studies presenting these instruments.
2) The values of the reliability coefficients should be accompanied by their respective confidence intervals.
3) Authors should be more critical in assessing the values of reliability coefficients. It is not credible that reliability values of .65 are considered adequate and that values of .97 also receive this assessment.
4) All reliability values should be given to the same number of decimal places.
5) Have the data had missing values? If so, how have they been treated?
6) In the mean comparisons (Table 1) there are statistically significant values at 5% (e.g. club-related stressors and education-related stressors) that are nevertheless interpreted as non-significant. Has another significance criterion been used?
7) The regression models explain an insignificant percentage of the variance. What value do the authors place on this result? How does this result affect the conclusions?
Author Response
Reviewer 1:
- The values of the reliability coefficients of the instruments used (lines 119. 130. and 142) should correspond to the data of the study. not to the data published in the studies presenting these instruments.
Thank you for pointing out that the reliability coefficients were not related to our study. We have revised and made changes. The newly presented reliability coefficiens are related to this study – lines 125, 136 and 148.
- The values of the reliability coefficients should be accompanied by their respective confidence intervals.
We have, accordingly, made and inserted a new table (Table 1.) which containes following data of the applied instruments - the means, standard deviations, minimum and maximum values, Cronbach’s αs. and confidence intervals of Cronbach’s - to emphasize this point. (line 172)
Table 1. Descriptive statistics and Cronbach alpha (N=1260)
|
Min. |
Max. |
M |
SD |
Cronbach α |
95% Confidence Interval |
|||
|
Lower Bound |
Upper Bound |
|||||||
|
PSSS |
Parental Stressors Total Score |
11 |
47 |
21.10 |
7.27 |
.82 |
.81 |
.84 |
|
Deselection |
4 |
20 |
7.64 |
3.38 |
.63 |
.6 |
.66 |
|
|
Club-Related Stressors |
3 |
12 |
5.87 |
2.36 |
.74 |
.72 |
.77 |
|
|
Feedback* |
2 |
10 |
4.82 |
2.64 |
.98* |
.97 |
.98 |
|
|
Education-Related Stressors |
2 |
10 |
2.77 |
1.47 |
.86 |
.85 |
.88 |
|
|
PISQ |
Directive Behavior |
6 |
30 |
14.56 |
5.85 |
.88 |
.87 |
.89 |
|
Parental Pressure |
4 |
20 |
6.70 |
2.96 |
.74 |
.71 |
.76 |
|
|
Active Involvement |
2 |
10 |
3.94 |
2.26 |
.78 |
.76 |
.80 |
|
|
Praise and Understanding |
5 |
15 |
13.93 |
1.58 |
.65 |
.62 |
.69 |
|
|
Perceived Parental Autonomy Support |
35 |
84 |
76.29 |
7.07 |
.83 |
.82 |
.85 |
|
Note. *the subscale contains two items; Apha values: 60<α<.70 acceptable; .70<α<.90 good (Taber, 2018)
- Authors should be more critical in assessing the values of reliability coefficients. It is not credible that reliability values of .65 are considered adequate and that values of .97 also receive this assessment.
Thank you for this observation. We have reformulated the reliability coefficients’ part at the end of the instruments section as follows (lines 125, 135 and 148).
We found the lower value of Praise and Understanding (Cronbach’s α=.65) and Deselection subscales (Cronbach’s α=.63) to be acceptable based on Taber (2018), and the value of .60-.70 indicates an acceptable level of reliability.
The original PISQ scale (Lee and McLean, 1998) has the same reliability value in the PU subscale (cr alpha = .60), and the spanish adaptation (Torregrosa és mtsai, 2005) was also close to the 0.70 limit (.72).
The Feedback subscale contains only two items - Correlations between items ( r=.64) increases the alpha value.
4) All reliability values should be given to the same number of decimal places.
Thank you for your comment. We agree with this and have incorporated your suggestion throughout the manuscript.
5) Have the data had missing values? If so, how have they been treated?
Thank you for your question. There were no missing data in our research’s database. We have noted the sample size in Table 1 (line 172).
6) In the mean comparisons (Table 1) there are statistically significant values at 5% (e.g. club-related stressors and education-related stressors) that are nevertheless interpreted as non-significant. Has another significance criterion been used?
Thank you very much for your clarifying comments. Mean differences were tested with robust statistical methods including Welch’s independent samples t-tests and Welch’s one-way ANOVA tests with Bonferroni correction for the subscales of PSSS – we have clarified and noted the value of Bonferroni adjustment (p< .013) on the lines 175-176.
7) The regression models explain an insignificant percentage of the variance. What value do the authors place on this result? How does this result affect the conclusions?
Thank you for the question. We made an effort to clarify the discussion part.
“Although the tested multiple linear regression proved significant for all types of parental stressors, the obtained coefficients of determination were small. Parenting is, of course, not confined to parents’ stress level or involvement in their children’s sport career, while the intensity of parental stressors itself may be influenced by a variety of factors. A useful interpretive framework is offered by Lazarus and Folkman’s [2] transactional model that defines stress as a relationship between the individual and their environment and assigns decisive importance to the individual’s perception of the situation in terms of the emergence of stress, and Bronfenbrenner’s ecological model [3,4], which points out the importance of the expectations held by the immediate environment and the norms prevailing in wider society in terms of parental stress. These factors fell outside the scope of the present study, while they might essentially contribute to the prediction of the intensity and types of parental subjective experience.” (lines 324-336)

Reviewer 2 Report
Exploring the relationship between children's participation in sports and parental stress is a very interesting topic, and the authors have studied this topic using questionnaires and drawn some conclusions.
There are six main aspects of our comments on this article:
1) There are two different relationships between children's athletic behavior and parental pressure: including the impact of children's athletic behavior on parental stress, the influence of parental stress on children's athletic behavior, and the selection of topics on the influence of parental pressure on children's athletic behavior is more meaningful than the impact of children's athletic behavior on parental stress, because children's participation in sports is the most important. This article only explores the impact of children's athletic behavior on parental stress, and its significance and value are not great. Therefore, it is hoped that in follow-up studies it will be possible to see how parental pressure is the movement behavior of children.
2) The preface does not clearly state some of the previous research results on the pressure of children's exercise on parents, and the relationship between parental stress and children's exercise has always been unclear.
3) The resulting part of the article lacks some logic. First, it should be stated how the age at which the child participates in sports, the different stages, the different programs, and whether there are sports injuries is stressful for parents. Which factors are most important between them? The article does not rank the above influencing factors. Second, it should be stated whether there are differences in parental pressure in terms of parental education level, parental personality, parental previous sports experience, etc.
4) The author chose 84 professional coaches and wondered what was the point of choosing them? It is also not specified in the text.
5) The discussion part is not explained in the order of the research results part, that is, it is discussed from two aspects: how the child's condition creates the pressure of parents, and whether the pressure is different among different parents, and provides evidence.
6) The conclusion part is unclear. Readers should be told how the child's exercise situation causes parental stress, rather than how parental pressure affects the child's exercise, which are two completely different issues and cannot be confused.
It is a great pity that the problems in the article are large, and I hope that the author will continue to work hard and strive for publication.
Author Response
Reviewer 2:
- There are two different relationships between children's athletic behavior and parental pressure: including the impact of children's athletic behavior on parental stress, the influence of parental stress on children's athletic behavior, and the selection of topics on the influence of parental pressure on children's athletic behavior is more meaningful than the impact of children's athletic behavior on parental stress, because children's participation in sports is the most important. This article only explores the impact of children's athletic behavior on parental stress, and its significance and value are not great. Therefore, it is hoped that in follow-up studies it will be possible to see how parental pressure is the movement behavior of children.
While we highly appreciate the reviewer’s honest feedback, and the thorough work they devoted to improving our manuscript, we would like to challenge these statements. We agree with the reviewer that the sport parenting is a minor field of the sport and exercise psychology, and the parental stress is a special area in this subfield.
Because of social and political impacts (i.e., politics, economical, and technological) youth sport is changing for participants (i.e., athletes, parents, coaches), where the expectations related to performance and competition turn out to be greater (Todd et al, 2020; Dorsch et al, 2021).
Several studies have found that not only coaches but also parents play a pivotal role in the development of athletes’ mindsets as the relationship between athletes and their parents’ attitudes towards sport sport (Anderson et al., 2003; Ullrich-French & Smith, 2006; Xiang et al., 2003), goal orientation (Atkins et al., 2015; Juntumaa, Keskivaara & Punamaki, 2005), motivation (Cope et al., 2013) and perfectionism (Sapieja et al., 2011) were detected. Studies have also emphasized the stressful nature of sport parenting and the negative effect this can have on their experiences (Burgess et al., 2016; Clarke and Harwood, 2012, Harwood & Knight, 2009; Harwood et al., 2010; Lienhart et al., 2020). Studies also highlighted an association between parental behaviour and the level of anxiety experienced by children (Kaye et al., 2014; Scanlan & Lewthwaite, 1984, 1986) as well as their self-confidence (Sorkkila, Aunola & Ryba, 2017).
The previous studies have predominantly focused on understanding the stressors parents face across different developmental stages by using qualitative method. The purpose of our research was to confirm existing theories related to parental stress in sport rather than to expanding them with quantitative methodology.
The complexity of the issue is demonstrated by the high number of studies focusing on the dyadic relationship of parent-athlete and attempts to provide an alternative solution for professionals. For instance, the methodology of family therapy based on a systemic approach (Zito, 2011) also highlights the direct effect of parental expectations, beliefs and behaviour on children's sport performance. Moreover, multiple psychoeducational books and programmes set out to help parents to create a suitable and effective atmosphere (Dorsch et al., 2016; Smoll, Smith & Cumming, 2007; Thrower, Harwood & Spray, 2017; Vincent & Christensen, 2015). Further development models also assign a key role to the parental background (Balyi, Higgs & Way, 2013; Fraser-Thomas, Côté & Deakin, 2015; Henriksen, Stambulova & Roessler, 2010).
Introduction:
„Because of social and political impacts (i.e., politics, economical, and technological) youth sport is changing for participants (i.e., athletes, parents, coaches), where the ex-pectations related to performance and competition turn out to be greater [5,6]. Several studies have found that not only coaches but also parents play a pivotal role in the de-velopment of athletes’ mindsets as the relationship between athletes and their parents’ attitudes towards sport [7-9], goal orientation [10-11], motivation [12] and perfection-ism [13] were detected. Studies have also emphasized the stressful nature of sport parenting and the negative effect this can have on their experiences [14-18]. According to qualitative and quantitative studies, sporting children’s parents face different stressors compared to other parents.” (lines 46-55)
- The preface does not clearly state some of the previous research results on the pressure of children's exercise on parents, and the relationship between parental stress and children's exercise has always been unclear.
Thank you for this observation. We reformulated the introduction parts and inserted the above-mentioned paragraph:
“Several studies have found that not only coaches but also parents play a pivotal role in the development of athletes’ mindsets as the relationship between athletes and their parents’ attitudes towards sport [7-9], goal orientation [10-11], motivation [12] and perfectionism [13] were detected. Studies have also emphasized the stressful nature of sport parenting and the negative effect this can have on their experiences [14-18]. According to qualitative and quantitative studies, sporting children’s parents face different stressors compared to other parents.” (lines 49-55)”
- The resulting part of the article lacks some logic. First, it should be stated how the age at which the child participates in sports, the different stages, the different programs, and whether there are sports injuries is stressful for parents.
Thank you for this feedback.
At first, our data was analyzed for possible mean differences on the five PSSS scales between groups based on variables tested in previous studies (the parent’s gender, the child’s gender, the parent’s previous sporting experience, the child’s sport participation stage) and on variables introduced in the present study (the child’s sport injuries, the parent’s marital status, the parent’s level of education, and the number of children in the family).
Which factors are most important between them? The article does not rank the above influencing factors.
The regression models explained a small significant percentage of the variance, thus it was irrelevant to rank influencing factors. Beta values of significant factors were nearly equal. We kept the above-mentioned line in the limitations section.
“Although the tested multiple linear regression proved significant for all types of parental stressors, the obtained coefficients of determination were small. Parenting is, of course, not confined to parents’ stress level or involvement in their children’s sport career, while the intensity of parental stressors itself may be influenced by a variety of factors. A useful interpretive framework is offered by Lazarus and Folkman’s [2] transactional model that defines stress as a relationship between the individual and their environment and assigns decisive importance to the individual’s perception of the situation in terms of the emergence of stress, and Bronfenbrenner’s ecological model [3,4], which points out the importance of the expectations held by the immediate environment and the norms prevailing in wider society in terms of parental stress. These factors fell outside the scope of the present study, while they might essentially contribute to the prediction of the intensity and types of parental subjective experience.” (lines 324-336)
Second, it should be stated whether there are differences in parental pressure in terms of parental education level, parental personality, parental previous sports experience, etc.
Thank you for your comment. The participants provided data on both their own and their children’s sociodemographic and sport-relevant characteristics (parents’ gender, age, educational level, and past sporting experiences; children’s gender, age, sport, training hours per week, active years in the sport, sports participation stage, and past injuries). No personality measure was employed in the present study, thus these relationships remain the subject of further research.
“The parents of children with a previous sports injury reported higher levels of fear of deselection and negative experiences related to communications from the club compared to the parents of children with no previous sports injury. Furthermore, the parents with tertiary education reported higher levels of concern about their children's academic progress compared to those with secondary education (see Table 3). No significant difference on either PSSS subscale was associated with the parent’s gender, the child’s gender, the parent’s previous sporting experience (yes/no), the parent’s marital status (single/married/cohabitant), or the number of children in the household (one/two/three or more).”
- The author chose 84 professional coaches and wondered what was the point of choosing them? It is also not specified in the text
“The sample included a total of N=1260 participating parents (399 males and 861 females) aged 29 to 74 years (Mage = 43.54, SD = 5.10).”
“The participants were recruited via convenience sampling. The participants were contacted directly or with the help of sports associations, sports clubs, a sports academy and professional coaches.”
- The discussion part is not explained in the order of the research results part, that is, it is discussed from two aspects: how the child's condition creates the pressure of parents, and whether the pressure is different among different parents, and provides evidence.
Thank you very much for your clarifying comments. We made an effort to clarify the discussion part. We have revised the second paragraph of the discussion in which we used incorrect terminology - the parental involvement is actually related to parental tasks and expertise which derived from the parents’ experiences and role conflicts. We’ve made the changes and clarified the terms in the whole manuscript, and noted the examined development stages related to our results (lines 246-288).
- The conclusion part is unclear. Readers should be told how the child's exercise situation causes parental stress, rather than how parental pressure affects the child's exercise, which are two completely different issues and cannot be confused.
We agree with the statement that parental stress affects child's exercise, and the child's exercise situation which causes parental stress, are two different topics. The aim of our study was to explore the relative importance of stressors potentially influencing the parenting behaviour of junior athletes’ parents and the associations between these stressors and various types of parental involvement. These findings might have important implications for parental psychoeducation as well as for the training of sports professionals (coaches, sports medicine physicians, rehabilitation experts).

Reviewer 3 Report
I believe this study is very well designed and has a merit for this journal. My only suggestion is about adding more up to date (2020 and above) literature review to current study. Best Regards.
Author Response
Reviewer 3.
I believe this study is very well designed and has a merit for this journal. My only suggestion is about adding more up to date (2020 and above) literature review to current study. Best Regards.
We appreciate your overview and summary of the manuscript. We incorporated four related articles into the text:
Todd, J., & Edwards, J. R. (2021). Understanding parental support in elite sport: A phenomenological approach to exploring midget triple a hockey in the Canadian Maritimes. Sport in Society, 24(9), 1590-1608. https://doi.org/10.1080/17430437.2020.1763311
Dorsch, T. E., Wright, E., Eckardt, V. C., Elliott, S., Thrower, S. N., & Knight, C. J. (2021). A history of parent involvement in organized youth sport: A scoping review. Sport, Exercise, and Performance Psychology, 10(4), 536–557 https://doi.org/10.1037/spy0000266
Bonavolontà, V., Cataldi, S., Latino, F., Carvutto, R., De Candia, M., Mastrorilli, G., ... & Fischetti, F. (2021). The Role of Parental Involvement in Youth Sport Experience: Perceived and Desired Behavior by Male Soccer Players. International Journal of Environmental Research and Public Health, 18(16), 8698. https://doi.org/10.3390/ijerph18168698
Prosoli, R., Lochbaum, M., & Barić, R. (2021). Parents at the sport competition: How they react, feel and cope with the event. Pedagogy of Physical Culture and Sports. 25(2):114-2. https://doi.org/10.15561/26649837.2021.0206

Round 2
Reviewer 2 Report
The author has revised the paper well, but it still needs to be improved.
Main suggestions: in the summary and conclusion part, it only explains the impact of sports injury on parents' pressure, but does not explain how the children's parents' education level has an impact on parents' pressure, and what stages of children's participation in sports have an impact on parents' pressure.
Author Response
Thank you for this observation. We made an effort to clarify the discussion part. We have reformulated the discussion parts and inserted these sentences:
“The results of the present study shows that the child’s sport participation stage could be a significant predictor of parental stress and confirm that their stress is lowest during the sampling years (under 12 years), while it substantially increases after the child reaches 12 years old when parental participation is significant in the child’s sport life.” (lines 246-249)
„This is in line with the findings of the present study in which the parents' educational level might predict their level of stress. Our results indicate that the parents with secondary (i.e., relatively low) education may have found communication with coaches and the club less stressful than those with tertiary education, who were more concerned about their children’s academic progress and more likely to confront the club over its communication.” (lines 293-298)
